# A Catalyst Framework for the Quantum Linear System Problem via the Proximal Point Algorithm

## Abstract

Solving systems of linear equations is a fundamental problem, but it can be computationally intensive for classical algorithms in high dimensions. Existing quantum algorithms can achieve exponential speedups for the quantum linear system problem (QLSP) in terms of the problem dimension, but even such a theoretical advantage is bottlenecked by the condition number of the coefficient matrix. In this work, we propose a new quantum algorithm for QLSP inspired by the classical proximal point algorithm (PPA). Our proposed method can be viewed as a meta-algorithm that allows inverting a modified matrix via an existing `QLSP_solver`, thereby directly approximating the solution vector instead of approximating the inverse of the coefficient matrix. By carefully choosing the step size $\eta$, the proposed algorithm can effectively precondition the linear system to mitigate the dependence on condition numbers that hindered the applicability of previous approaches. Importantly, this is the first framework for QLSP where a tunable parameter $\eta$ allows the user to control the trade-off between the runtime and the approximation error.

## 1 Introduction

**Background.** Solving systems of linear equations is a fundamental problem with many applications spanning science and engineering. Mathematically, for a given (Hermitian) matrix $A \in \mathbb{C}^{N \times N}$ and vector $b \in \mathbb{C}^N$, the goal is to find the $N$-dimensional vector $x^\star$ that satisfies $Ax^\star = b$. While classical algorithms, such as Gaussian elimination (Gauss, 1877; Higham, 2011), conjugate gradient method (Hestenes & Stiefel, 1952), $LU$ factorization (Schwarzenberg-Czerny, 1995; Shabat et al., 2018), $QR$ factorization (Francis, 1961; Kublanovskaya, 1962), and iterative Krylov subspace methods (Krylov, 1931; Nocedal & Wright, 1999), can solve this problem, their complexity scales (at worst) cubically with $N$, the dimension of $A$, motivating the development of quantum algorithms that could potentially achieve speedups.

Indeed, for the quantum linear system problem (QLSP) in Definition 1,[1] Harrow, Hassidim, and Lloyd (a.k.a. the HHL algorithm) (Harrow et al., 2009) showed that the dependence on the problem dimension exponentially reduces to $\mathcal{O}\left(\text{poly} \log(N)\right)$, with query complexity of $\mathcal{O}(\kappa^2/\varepsilon)$, under some (quantum) access model for $A$ and $b$ (c.f., Definitions 2 and 3). Here, $\kappa$ is the condition number of $A$, defined as the ratio of the largest to the smallest singular value of $A$. Subsequent works, such as the work by Ambainis (Ambainis, 2012) and by Childs, Kothari, and Somma (Childs et al., 2017) (a.k.a. the CKS algorithm) improve the dependence on $\kappa$ and $\varepsilon$; see Table 1. The best quantum algorithm for QLSP is based on the discrete adiabatic theorem (Costa et al., 2022), achieving the query complexity of $\mathcal{O}(\kappa \cdot \log(1/\varepsilon))$, matching the lower bound (Orsucci & Dunjko, 2021).[2]

Solving QLSP is a fundamental subroutine in many quantum algorithms. For instance, it is used in quantum recommendation systems (Kerenidis & Prakash, 2016), quantum SVM (Rebentrost et al., 2014), unsupervised learning (Wiebe et al., 2014), and solving differential equations (Liu et al., 2021), to name a few. Hence, improving the overall runtime of a generic `QLSP_solver` is crucial in developing more sophisticated and efficient quantum algorithms.

---

[1] Note that QLSP is BQP-complete (Dervovic et al., 2018; Prasad & Zhuang, 2022).

[2] This also matches the iteration complexity of the (classical) conjugate gradient method (Hestenes & Stiefel, 1952).

**Challenges in existing methodologies.** A common limitation of the existing quantum algorithms is that the dependence on the condition number $\kappa$ must be small to achieve the quantum advantage. To put more context, for any quantum algorithm in Table 1 to achieve an exponential advantage over classical algorithms, $\kappa$ needs to be in the order of poly $\log(N)$, where $N$ is the dimension of $A$. For instance, for $N = 10^3$, $\kappa$ needs to be around 4 to exhibit the exponential advantage. However, condition numbers are often large in real-world problems (Papyan, 2020).

Moreover, it was proven (Orsucci & Dunjko, 2021, Proposition 6) that for QLSP, even when $A$ is *positive-definite*, the dependence on condition number *cannot* be improved from $\mathcal{O}(\kappa)$. This is in contrast to the classical algorithms, such as the conjugate gradient method, which achieves $\mathcal{O}(\sqrt{\kappa})$ reduction in complexity for the linear systems of equations with $A \succ 0$. Such observation reinforces the importance of alleviating the dependence on the condition number $\kappa$ for *quantum algorithms*, which is our aim.

**Our contributions.** We present a novel meta-algorithm for solving the QLSP based on the proximal point algorithm (PPA) (Rockafellar, 1976; Güler, 1991); see Algorithm 1. Notably, in contrast to some existing methods that approximate $A^{-1}$ (Harrow et al., 2009; Ambainis, 2012; Childs et al., 2017; Gribling et al., 2021; Orsucci & Dunjko, 2021), our algorithm directly approximates $x^\star = A^{-1}b$ through an iterative process based on PPA. Classically, PPA is known to improve the "conditioning" of the problem at hand, compared to the gradient descent (Toulis et al., 2014; Toulis & Airoldi, 2017; Ahn & Sra, 2022); it can also be accelerated (Güler, 1992; Kim et al., 2022).

An approximate proximal point algorithm for classical convex optimization has been proposed under the name of "catalyst" (Lin et al., 2015) in the machine learning community. Our proposed method operates similarly and can be viewed as a generic acceleration scheme for QLSP where one can plug in different `QLSP_solver` –e.g., HHL, CKS– to achieve generic (constant-level) acceleration. In Figure 1, we illustrate the case where the best quantum algorithm for QLSP (Costa et al., 2022), based on the discrete adiabatic theorem, is utilized as a subroutine for Algorithm 1. The improvement in the query complexity compared to the original algorithm to achieve a fixed accuracy increases as $\kappa$ increases.

Intuitively, by the definition of PPA detailed in Section 3, our proposed method allows to invert a modified matrix $I + \eta A$ and arrive at the same solution $A^{-1}b$, as shown below:

$$x_{t+1} = (I + \eta A)^{-1}(x_t + \eta b) \underset{\eta \to \infty}{=} A^{-1}b.$$

Yet, the key feature and the main distinction here is the introduction of a *tunable (step size) parameter* $\eta$ that allows pre-conditioning the linear system. By carefully choosing $\eta$, we can invert the modified matrix $I + \eta A$ that is better conditioned than $A$, thereby mitigating the dependence on $\kappa$ of the existing quantum algorithms; see also Remark 3 and Figure 1.

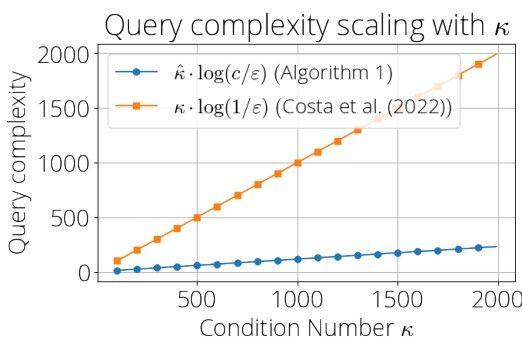

Figure 1: *Query complexity scaling with respect to the condition number $\kappa$.* Here, the best quantum algorithm (Costa et al., 2022) is used as `QLSP_solver` for the subroutine in Algorithm 1. Simply by "wrapping" the `QLSP_solver`, one can achieve much better scaling with respect to the condition number $\kappa$.

Our meta-algorithmic framework complements and provides advantages over prior works on QLSP. Most importantly, it alleviates the strict requirements on the condition number, enabling quantum speedups for a broader class of problems where $\kappa$ may be large. Moreover, the $\eta$ parameter can be tuned to balance the convergence rate and precision demands, providing greater flexibility in optimizing the overall algorithmic complexity.

In summary, our contributions and findings are as follows:

- We propose a meta-algorithmic framework for the quantum linear system problem (QLSP) based on the proximal point algorithm. Unlike existing quantum algorithms for QLSP, which rely on different unitary approximations of $A^{-1}$, our proposed method allows to invert a modified matrix with a smaller condition number (c.f., Remark 3 and Lemma 1), when $A$ is positive-definite.

- To our knowledge, this is the first framework for QLSP where a *tunable parameter* $\eta$ allows the user to control the trade-off between the runtime (query complexity) and the approximation error (solution quality). Importantly, there exist choices of $\eta > 0$ that allow to decrease the runtime while maintaining the same error level (c.f., Theorem 3).

- Our proposed method allows to achieve significant constant-level improvements in the query complexity, even compared to the best quantum algorithm for QLSP (Costa et al., 2022), simply by using it as a subroutine of Algorithm 1. This is possible as the improvement "grows faster" than the overhead (c.f., Figure 2 and Theorem 6).

## 2 PROBLEM SETUP AND RELATED WORK

**Notation.** Matrices are represented with uppercase letters as in $A \in \mathbb{C}^{N \times N}$; vectors are represented with lowercase letters as in $b \in \mathbb{C}^N$, and are distinguished from scalars based on the context. The condition number of a matrix $A$, denoted as $\kappa$, is the ratio of the largest to the smallest singular value of $A$. We denote $\|\cdot\|$ as the Euclidean $\ell_2$-norm. *Qubit* is the fundamental unit in quantum computing, analogous to the bit in classical computing. The state of a qubit is represented using the bra-ket notation, where a single qubit state $|\psi\rangle \in \mathbb{C}^2$ can be expressed as a linear combination of the basis states $|0\rangle = [0 \ 1]^\top \in \mathbb{C}^2$ and $|1\rangle = [1 \ 0]^\top \in \mathbb{C}^2$, as in $|\psi\rangle = \alpha|0\rangle + \beta|1\rangle$; here, $\alpha, \beta \in \mathbb{C}$ are called *amplitudes* and encode the probability of the qubit collapsing to either, such that $|\alpha|^2 + |\beta|^2 = 1$. Thus, $|\psi\rangle$ represents a normalized column vector by definition. $|\cdot\rangle$ is a column vector (called *bra*), and its conjugate transpose (called *ket*), denoted by $\langle\cdot|$, is defined as $\langle\cdot| = |\cdot\rangle^*$. Generalizing the above, an $n$-qubit state is a unit vector in $n$-qubit Hilbert space, defined as the Kronecker product of $n$ single qubit states, i.e., $\mathcal{H} = \otimes_{i=1}^n \mathbb{C}^2 \cong \mathbb{C}^{2^n}$. It is customary to write $2^n = N$. Quantum states can be manipulated using quantum gates, represented by unitary matrices that act on the state vectors. For example, a single-qubit gate $U \in \mathbb{C}^{2 \times 2}$ acting on a qubit state $|\psi\rangle$ transforms it to $U|\psi\rangle$, altering the state's probability amplitudes according to the specific operation represented by $U$.

### 2.1 THE QUANTUM LINEAR SYSTEM PROBLEM

In the quantum setting, the goal of the quantum linear system problem (QLSP) is to prepare a *quantum state* proportional to the vector $x^\star$. That is, we want to output $|x^\star\rangle := \frac{\sum_i x_i^\star |i\rangle}{\|\sum_i x_i^\star |i\rangle\|}$ where the vector $x^\star = [x_1^\star, \ldots, x_N^\star]^\top$ satisfies $Ax^\star = b$. Formally, we define the QLSP problem as below.

**Definition 1** (Quantum Linear System Problem (Childs et al., 2017)). *Let $A$ be an $N \times N$ Hermitian matrix satisfying $\|A\| = 1$ with condition number $\kappa$ and at most $s$ nonzero entries in any row or column. Let $b$ be an $N$-dimensional vector, and let $x^\star := A^{-1}b$. We define the quantum states $|b\rangle$ and $|x^\star\rangle$ as in:*

$$|b\rangle := \frac{\sum_{i=0}^{N-1} b_i |i\rangle}{\left\|\sum_{i=0}^{N-1} b_i |i\rangle\right\|} \quad and \quad |x^\star\rangle := \frac{\sum_{i=0}^{N-1} x_i^\star |i\rangle}{\left\|\sum_{i=0}^{N-1} x_i^\star |i\rangle\right\|}. \tag{1}$$

*Given access to $A$ via $\mathcal{P}_A$ in Definition 3 or $\mathcal{U}_A$ in Definition 4, and access to the state $|b\rangle$ via $\mathcal{P}_B$ in Definition 2, the goal of QLSP is to output a state $|\tilde{x}\rangle$ such that $\||\tilde{x}\rangle - |x^\star\rangle\| \leqslant \varepsilon$.*

As in previous works (Harrow et al., 2009; Childs et al., 2017; Ambainis, 2012; Gribling et al., 2021; Orsucci & Dunjko, 2021; Costa et al., 2022), we assume that access to $A$ and $b$ is provided by black-box subroutines that we detail below. We start with the state preparation oracle for the vector $b \in \mathbb{C}^N$.

**Definition 2** (State preparation oracle (Harrow et al., 2009)). *Given a vector $b \in \mathbb{C}^N$, there exists a procedure $\mathcal{P}_B$ that prepares the state $|b\rangle := \frac{\sum_i b_i |i\rangle}{\|\sum_i b_i |i\rangle\|}$ in time $\mathcal{O}(poly \log(N))$.*

We assume two encoding models for $A$: the sparse-matrix-access in Definition 3 denoted by $\mathcal{P}_A$, and the matrix-block-encoding model (Gilyén et al., 2019; Low & Chuang, 2019) in Definition 4, denoted by $\mathcal{U}_A$, respectively.

**Definition 3** (Sparse matrix access (Childs et al., 2017)). *Given a $N \times N$ Hermitian matrix $A$ with operator norm $\|A\| \leqslant 1$ and at most $s$ nonzero entries in any row or column, $\mathcal{P}_A$ allows the following*

| QLSP_solver | Query Complexity | Key Technique/Result |
|---|---|---|
| HHL (Harrow et al., 2009) | $\mathcal{O}\left(\kappa^2/\varepsilon\right)$ | First quantum algorithm for QLSP |
| Ambainis (Ambainis, 2012) | $\mathcal{O}\left(\kappa \log^3(\kappa)/\varepsilon^3\right)$ | Variable Time Amplitude Amplification |
| CKS (Childs et al., 2017) | $\mathcal{O}\left(\kappa \cdot \operatorname{poly}\log(\kappa/\varepsilon)\right)$ | (Truncated) Chebyshev bases via LCU |
| Subaşı et al. (Subaşı et al., 2019) | $\mathcal{O}\left((\kappa \log \kappa)/\varepsilon)\right)$ | Adiabatic Randomization Method |
| An & Lin (An & Lin, 2022) | $\mathcal{O}\left(\kappa \cdot \operatorname{poly}\log(\kappa/\varepsilon)\right)$ | Time-Optimal Adiabatic Method |
| Lin & Tong (Lin & Tong, 2020) | $\mathcal{O}\left(\kappa \cdot \log(\kappa/\varepsilon)\right)$ | Zeno Eigenstate Filtering |
| Costa, et al. (Costa et al., 2022) | $\mathcal{O}\left(\kappa \cdot \log(1/\varepsilon)\right)$ | Discrete Adiabatic Theorem |
| **Ours** | $\kappa \to \frac{\kappa(1+\eta)}{\kappa+\eta}$ for all above | Proximal Point Algorithm |

Table 1: *Query complexities and key results used in related works on QLSP.* Our proposed framework in Algorithm 1 allows to improve the dependence on the condition number $\kappa$ for any QLSP_solver, so long as the input matrix $A$ is positive-definite.

*mapping:*

$$|j,\ell\rangle \mapsto |j,\nu(j,\ell)\rangle \qquad \forall j \in [N] \text{ and } \ell \in [s], \qquad (2)$$

$$|j,k,z\rangle|0\rangle \mapsto |j,k,z \oplus A_{jk}\rangle \qquad \forall j,k \in [N], \qquad (3)$$

*where $\nu : [N] \times [s] \to [N]$ in (2) computes the row index of the $\ell$th nonzero entry of the $j$th column, and the third register of (3) holds a bit string representing the entry of $A_{jk}$.*

**Definition 4** (Matrix block-encoding (Gilyén et al., 2019)). *A unitary operator $\mathcal{U}_A$ acting on $n + c$ qubits is called an $(\alpha, c, \varepsilon)$-matrix-block-encoding of a $n$-qubit operator $A$ if*

$$\|A - \alpha((\langle 0^c| \otimes I) \mathcal{U}_A (|0^c\rangle \otimes I))\| \leqslant \varepsilon.$$

*The above can also be expressed as follows:*

$$\mathcal{U}_A = \begin{bmatrix} \tilde{A}/\alpha & * \\ * & * \end{bmatrix} \quad \text{with} \quad \|\tilde{A} - A\| \leqslant \varepsilon,$$

*where $*$'s denote arbitrary matrix blocks with appropriate dimensions.*

In (Childs et al., 2017), it was shown that a $(s, 1, 0)$-matrix-block-encoding of $A$ is possible using a constant number of calls to $\mathcal{P}_A$ in Definition 3 (and $\mathcal{O}(\operatorname{poly}(n))$ extra elementary gates). In short, $\mathcal{P}_A$ in Definition 3 implies efficient implementation of $\mathcal{U}_A$ in Definition 4 (c.f., (Gilyén et al., 2019, Lemma 48)). We present both for completeness as different works rely on different access models; however, our proposed meta-algorithm can provide generic acceleration for any QLSP solver, regardless of the encoding method.

## 2.2 RELATED WORK

**Quantum algorithms.** We summarize the related quantum algorithms for QLSP and their query complexities in Table 1; all QLSP_solvers share the exponential improvement on the input dimension, $\mathcal{O}(\operatorname{poly}\log(N))$. The HHL algorithm (Harrow et al., 2009) utilizes quantum subroutines including $(i)$ Hamiltonian simulation (Feynman, 1982; Lloyd, 1996; Childs et al., 2018) that applies the unitary operator $e^{iAt}$ to $|b\rangle$ for a superposition of different times $t$, $(ii)$ phase estimation (Kitaev, 1995) that allows to decompose $|b\rangle$ into the eigenbasis of $A$ and to find its corresponding eigenvalues, and $(iii)$ amplitude amplification (Brassard & Hoyer, 1997; Grover, 1998; Brassard et al., 2002) that allows to implement the final state with amplitudes the same with the elements of $x^\star$. Subsequently, (Ambainis, 2012) achieved a quadratic improvement on the condition number at the cost of worse error dependence; the main technical contribution was to improve the amplitude amplification, the previous bottleneck. CKS (Childs et al., 2017) significantly improved the suboptimality by the linear combination of unitaries. We review these subroutines in the appendix.

(Costa et al., 2022) is the state-of-the-art QLS algorithm based on the adiabatic framework, which was spearheaded by (Subaşı et al., 2019) and improved in (An & Lin, 2022; Lin & Tong, 2020). These are significantly different from the aforementioned HHL-based approaches. Importantly, Algorithm 1 is oblivious to such differences and provides generic acceleration.

**Lower bounds.** Along with the proposal of the first quantum algorithm for QLSP, (Harrow et al., 2009) also proved the lower bound of $\Omega(\kappa)$ queries to the entries of the matrix is needed for general linear systems. In (Orsucci & Dunjko, 2021, Proposition 6), this lower bound was surprisingly extended to the case of *positive-definite* systems. This is in contrast to the classical optimization literature, where methods such as the conjugate gradient method (Hestenes & Stiefel, 1952) achieve $\sqrt{\kappa}$-acceleration for positive-definite systems.

# 3 THE PROXIMAL POINT ALGORITHM FOR QLSP

We now introduce our proposed methodology, summarized in Algorithm 1. At a high level, our method can be viewed as a meta-algorithm, where one can plug in any existing QLSP_solver as a subroutine to achieve generic acceleration. We first review the proximal point algorithm (PPA), a classical optimization method on which Algorithm 1 is based.

## 3.1 THE PROXIMAL POINT ALGORITHM

We take a step back from the QLSP in Definition 1 and introduce the proximal point algorithm (PPA), a fundamental optimization method in convex optimization (Rockafellar, 1976; Güler, 1991; Parikh et al., 2014; Bauschke & Combettes, 2019). PPA is an iterative algorithm that proceeds by minimizing the original function plus an additional quadratic term, as in:

$$x_{t+1} = \arg\min_x \left\{ f(x) + \frac{1}{2\eta} \|x - x_t\|_2^2 \right\}. \tag{4}$$

As a result, it changes the "conditioning" of the problem; if $f(\cdot)$ is convex, the optimization problem in (4) can be strongly convex (Ahn & Sra, 2022). By the first-order optimality condition (Boyd & Vandenberghe, 2004, Eq. (4.22)), (4) can be written in the following form, also known as the implicit gradient descent (IGD):

$$x_{t+1} = x_t - \eta \nabla f(x_{t+1}). \tag{5}$$

(5) is an implicit method and generally cannot be implemented. However, the case we are interested in is the quadratic minimization problem:

$$\min_x f(x) = \frac{1}{2} x^\top A x - b^\top x. \tag{6}$$

Then, we have the closed-form update for (5) as follows:

$$x_{t+1} = x_t - \eta(Ax_{t+1} - b) = x_t - \eta A(x_{t+1} - x^\star).$$

In particular, by rearranging and unfolding, we have

$$x_{t+1} = (I + \eta A)^{-1}(x_t + \eta b) = \cdots = (I + \eta A)^{-(t+1)} x_0 + \eta b \sum_{k=1}^{t+1} (I + \eta A)^{-k}. \tag{7}$$

The above expression sheds some light on how applying PPA can differ from simple inversion: $A^{-1}b$. In particular, PPA enables to invert a modified matrix $I + \eta A$ based on $\eta$; see also Remark 3 below.

Further, since $b = Ax^\star$, we can equivalently express the series of operations in (7) as follows:

$$x_{t+1} - x^\star = (I + \eta A)^{-1}(x_t - x^\star) = \cdots = (I + \eta A)^{-t}(x_0 - x^\star). \tag{8}$$

The above expression helps compute the number of iterations required for PPA, given $\eta > 0$, for finding $\varepsilon$-approximate solution, as we detail in Section 4.

## 3.2 META-ALGORITHM FOR QLSP VIA PPA

We present our proposed method, which is extremely simple as summarized in Algorithm 1.

Line 2 is the cornerstone of the algorithm where one can employ any QLSP_solver –like HHL (Harrow et al., 2009), CKS (Childs et al., 2017) or the recent work based on discrete adiabatic approach (Costa et al., 2022)– to the (normalized) matrix $(I + \eta A)/\|I + \eta A\|$, enabled by the PPA approach. In other words, line 3 can be seen as the output of applying QLSP_solver$(\frac{I+\eta A}{\|I+\eta A\|}, |x_0 + \eta b\rangle)$. We make some remarks on the input.

---

**Algorithm 1** Proximal Point Algorithm for the Quantum Linear System Problem

---

1: **Input**: An oracle $\mathcal{P}_{\eta,A}$ that prepares sparse-access or block-encoding of $\frac{I+\eta A}{\|I+\eta A\|}$ (c.f., Definition 3 and Definition 4); a state preparation oracle $\mathcal{P}_{b,x_0,\eta}$ that prepares $|x_0 + \eta b\rangle$ (c.f., Definition 2); and a tunable step size $\eta > 0$.

2: **Subroutine:** Invoke any QLSP_solver such that $\left|\psi_{I+\eta A, x_0+\eta b}\right\rangle \approx \left|\left(\frac{I+\eta A}{\|I+\eta A\|}\right)^{-1}(x_0 + \eta b)\right\rangle$.

3: **Output:** Normalized quantum state $\left|\psi_{I+\eta A, x_0+\eta b}\right\rangle$.

4: **Benefit:** Improved dependence on condition number as in Remark 3.

---

**Remark 1** (Access to $(I + \eta A)/\|I + \eta A\|$). *Our algorithm necessitates an oracle $P_{\eta,A}$, which can provide either sparse access to $(I + \eta A)/\|I + \eta A\|$ (as in Definition 3) or its block encoding (as in Definition 4). For sparse access to $(I + \eta A)/\|I + \eta A\|$, direct access is feasible from sparse access to $A$, when all diagonal entries of $A$ are non-zero. Specifically, the access described in (3) is identical to that of $A$, while the access in (2) modifies $A_{jj}$ to $(1 + \eta A_{jj})/\|I + \eta A\|$. If $A$ contains zero diagonal entries, sparse access to $(I + \eta A)/\|I + \eta A\|$ requires only two additional uses of (3).*

*For block encoding, $(I + \eta A)/\|I + \eta A\|$ can be achieved through a linear combination of block-encoded matrices, as demonstrated in (Gilyén et al., 2019, Lemma 52). This method allows straight-forward adaptation of existing data structures that facilitate sparse-access or block-encoding of $A$ to also support $(I + \eta A)/\|I + \eta A\|$.*

*In addition, original data or data structures that provide sparse access or block-encoding of $A$ can be easily modified for access to $(I + \eta A)/\|I + \eta A\|$. For instance, suppose that we are given a matrix as a sum of multiple small matrices as in the local Hamiltonian problem or Hamiltonian simulation (see (Nielsen & Chuang, 2001) for an introduction). Then the sparse access and block encoding of both $A$ and $(I + \eta A)/\|I + \eta A\|$ can be efficiently derived from the sum of small matrices.*

**Remark 2** (Access to $|x_0 + \eta b\rangle$). *We prepare the initial state $|x_0 + \eta b\rangle$, instead of $|b\rangle$, reflecting the PPA update in (7). For this step, we assume there exists an oracle $\mathcal{P}_{b,x_0,\eta}$ such that $|x_0 + \eta b\rangle$ can be efficiently prepared similarly to Definition 2; otherwise, we can simply initialize $x_0$ with zero vector.*

Since we are inverting the (normalized) modified matrix $\frac{I+\eta A}{\|I+\eta A\|}$, the spectrum changes as follows.

**Lemma 1.** *Let $A$ be an $N \times N$ Hermitian positive-definite matrix satisfying $\|A\| = 1$ with condition number $\kappa$. Then, the condition number of the modified matrix in Algorithm 1, $\frac{I+\eta A}{\|I+\eta A\|}$, is given by $\hat{\kappa} = \frac{\kappa(1+\eta)}{\kappa+\eta}$.*

Notice that the modified condition number $\hat{\kappa}$ depends both on $\kappa$ and the step size parameter $\eta$ for PPA. As a result, $\eta$ plays a crucial role in the overall performance of Algorithm 1. The introduction of the tunable parameter $\eta$ in the context of QLSP is a main property that differentiates Algorithm 1 from other quantum algorithms. We summarize the trade-off of $\eta$ in the following remark.

**Remark 3.** *The modified condition number, $\hat{\kappa} = \frac{\kappa(1+\eta)}{\kappa+\eta}$, in conjunction with the PPA convergence in (8) introduces a trade-off based on the tunable parameter $\eta$.*

- *Large $\eta$ regime: setting large $\eta$ allows PPA to converge fast, as can be seen in (15). On the other hand, the benefit of the modified condition number diminishes and recovers the original $\kappa$:*

$$\hat{\kappa} = \frac{\kappa(1+\eta)}{\kappa+\eta} \xrightarrow[\eta\to\infty]{} \kappa,$$

- *Small $\eta$ regime: setting small $\eta$ slows down the convergence rate of PPA and requires more number of iterations, as can be seen in (15). On the other hand, the modified condition number becomes increasingly better conditioned, as in:*

$$\hat{\kappa} = \frac{\kappa(1+\eta)}{\kappa+\eta} \xrightarrow[\eta\to 0]{} 1.$$

# 4 THEORETICAL ANALYSIS

As shown in Algorithm 1 as well as the PPA iteration explained in (7), the main distinction of our proposed method from the existing QLSP algorithms is that we invert the modified matrix $(I + \eta A)/\|I + \eta A\|$ instead of the original matrix $A$. Precisely, our goal is to bound the following:

$$\left\| \underbrace{\left| \psi_{I+\eta A, x_0 + \eta b} \right\rangle - \left| \left( \tfrac{I+\eta A}{\|I+\eta A\|} \right)^{-1} (x_0 + \eta b) \right\rangle}_{\text{QLSP\_solver error (c.f., Proposition 1)}} + \underbrace{\left| \left( \tfrac{I+\eta A}{\|I+\eta A\|} \right)^{-1} (x_0 + \eta b) \right\rangle - \left| A^{-1}b \right\rangle}_{\text{PPA error (c.f., Proposition 2)}} \right\| \leqslant \varepsilon, \quad (9)$$

where $\left| \psi_{I+\eta A, x_0+\eta b} \right\rangle$ is the output of Algorithm 1, and $|A^{-1}b\rangle = \frac{A^{-1}b}{\|A^{-1}b\|}$ is the target quantum state of QLSP based on Definition 1. In between, the term $\left| \left( \tfrac{I+\eta A}{\|I+\eta A\|} \right)^{-1} (x_0 + \eta b) \right\rangle$ is added and subtracted, reflecting the modified inversion due to PPA. Specifically, the first pair of terms quantifies the error coming from the inexactness of any QLSP\_solver used as a subroutine of Algorithm 1. The second pair of terms quantifies the error coming from PPA in estimating $A^{-1}b$, as can be seen in (8). In the following subsections, we analyze each term carefully. Due to space limitations, we defer all proofs to the appendix at the end of the paper.

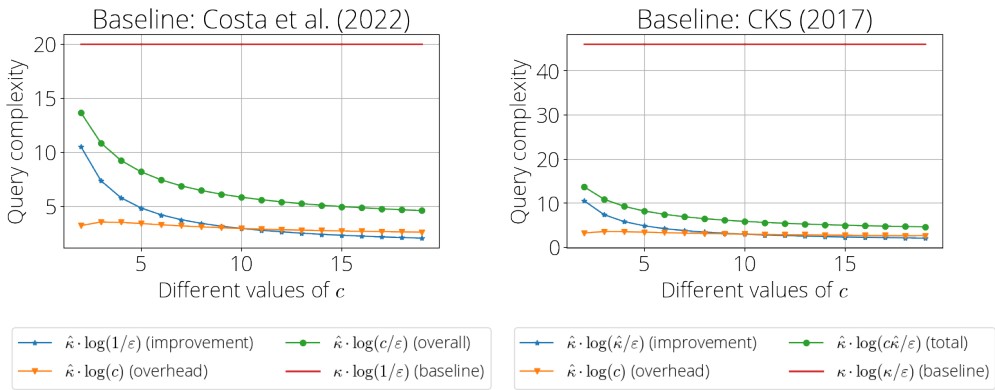

Figure 2: *How the "improvement" term, the "overhead" term, and their sum "total" allows improvement compared to the baselines from (Childs et al., 2017) and (Costa et al., 2022) (c.f., Theorem 6). Right panel: using CKS (Childs et al., 2017) as baseline (c.f., Theorem 4); Left panel: using (Costa et al., 2022) as baseline (c.f., Theorem 5). The key insight is that the rate of "improvement" is faster than the rate of "overhead," which plateaus quickly thanks to the logarithm as can be seen in Theorem 6.*

## 4.1 INVERTING THE MODIFIED MATRIX $(I + \eta A)/\|I + \eta A\|$

In the first part, we apply any QLSP\_solver to invert the modified matrix $(I + \eta A)/\|I + \eta A\|$. Importantly, we first need to encode it into a quantum computer, and this requires normalization so that the resulting encoded matrix is unitary, which is necessary for any quantum computer operation (c.f., Remarks 1 and 2). As in this part we are simply invoking existing QLSP\_solver, we just have to control the error $\varepsilon_1$ and its contribution to the final accuracy $\varepsilon$, as summarized below.

**Proposition 1** (QLSP\_solver error). *Assume access to the oracle $\mathcal{P}_{\eta,A}$ in Algorithm 1, similarly to the Definition 3 or 4 (c.f., Remark 1). Further, assume access to the oracle $\mathcal{P}_{b,x_0,\eta}$ similarly to the Definition 2 (c.f., Remark 2), respectively. Then, there exists quantum algorithms, such as the ones in Table 1, satisfying*

$$\left\| \left| \psi_{I+\eta A, x_0+\eta b} \right\rangle - \left| \left( \tfrac{I+\eta A}{\|I+\eta A\|} \right)^{-1} (x_0 + \eta b) \right\rangle \right\| \leqslant \varepsilon_1. \quad (10)$$

**Example 1** (CKS polynomial (Childs et al., 2017)). *CKS performed the following polynomial approximation of $A^{-1}$:*

$$A^{-1} \approx P(A) = \sum_i \alpha_i \mathcal{T}_i(A), \quad (11)$$

*where $\mathcal{T}_i$ denotes the Chevyshev polynomials (of the first kind). Then, given that $\|P(A) - A^{-1}\| \leqslant \varepsilon_1 < \frac{1}{2}$, (Childs et al., 2017, Proposition 9) proves the following:*

$$\left\| \frac{P(A)|\phi\rangle}{\|P(A)|\phi\rangle\|} - \frac{A^{-1}|\phi\rangle}{\|A^{-1}|\phi\rangle\|} \right\| \leqslant 4\varepsilon_1. \tag{12}$$

**Adapting Example 1 to Algorithm 1.** The approximation in (11) is only possible when the norm of the matrix to be inverted is upper-bounded by 1. Hence, we can use the following approximation:

$$\|I + \eta A\|^{-1} \sum_i \alpha_i \mathcal{T}_i \left( \frac{I + \eta A}{\|I + \eta A\|} \right) = \|I + \eta A\|^{-1} \cdot P \left( \frac{I + \eta A}{\|I + \eta A\|} \right), \tag{13}$$

which allows similar step as (12) to hold, and further allows the dependence on $\kappa$ for the CKS algorithm to be alleviated as summarized in Lemma 1; see also the illustration in Figure 2 (right).

## 4.2 Approximating $x^\star = A^{-1}b$ via PPA

For the PPA error (second pair of terms in (9)), the step size $\eta$ needs to be properly set up so that a single step of PPA, i.e., $x_1 = (I + \eta A)^{-1}(x_0 + \eta b)$ is close enough to $A^{-1}b = x^\star$. To achieve that, observe from (8):

$$\|x_{t+1} - x^\star\| \leqslant \|(I + \eta A)^{-t}\| \cdot \|x_0 - x^\star\| \leqslant \frac{1}{(1 + \eta\sigma_{\min})^t} \|x_0 - x^\star\|, \tag{14}$$

where $\sigma_{\min}$ is the smallest singular value of $A$. We desire the RHS to be less than $\varepsilon_2$. Denoting $\|x_0 - x^\star\| := d$ and using $\sigma_{\min} = 1/\kappa$ (c.f., Definition 1), we can compute the lower bound on the number of iterations $t$ as:

$$\frac{d}{(1 + \eta/\kappa)^t} \leqslant \varepsilon_2 \implies \frac{\log(\frac{d}{\varepsilon_2})}{\log(1 + \frac{\eta}{\kappa})} \leqslant t. \tag{15}$$

Based on the above analysis, we can compute the number of iterations $t$ required to have $\varepsilon$-optimal solution of the quadratic problem in (6). Here, (15) is well defined in the sense that the lower bound on $t$ is positive, as long as $\eta > 0$. In other words, if the LHS of (15) is less than 1, that means PPA converges to $\varepsilon_2$-approximate solution in one step, with a proper step size.

Again, our goal is to achieve (9). Hence, we have to characterize how (14) results in the proximity in corresponding normalized quantum states. We utilize Lemma 2 below.

**Lemma 2.** *Let $x$ and $y$ be two vectors. Suppose $\|x - y\| \leqslant \epsilon$ for some small positive scalar $\epsilon$. Then, the distance between the normalized vectors satisfies the following:*

$$\left\| \frac{x}{\|x\|} - \frac{y}{\|y\|} \right\| \leqslant \frac{\epsilon}{\sqrt{\|x\| \cdot \|y\|}}.$$

Now, we characterize the error of the normalized quantum state of as an output of PPA.

**Proposition 2.** *Running the PPA in (7) for a single iteration with $\eta = \kappa \left( \frac{d}{\varepsilon_2} - 1 \right)$, where $d := \|x_0 - A^{-1}b\|$, results in the normalized quantum state satisfying:*

$$\left\| \left| \left( \frac{I + \eta A}{\|I + \eta A\|} \right)^{-1}(x_0 + \eta b) \right\rangle - \left| A^{-1}b \right\rangle \right\| \leqslant \frac{\varepsilon_2}{\Psi}, \tag{16}$$

*where $\Psi := \sqrt{\|(I + \eta A)^{-1}(x_0 + \eta b)\| \cdot \|A^{-1}b\|}$.*

## 4.3 Overall Complexity and Improvement

Equipped with Propsitions 1 and 2, we arrive at the following theorem.

**Theorem 3** (Main result)**.** *Consider solving the QLSP problem in Definition 1 with Algorithm 1, of which the approximation error $\varepsilon$ can be decomposed as (9), recalled below:*

$$\left\| \underbrace{\left| \psi_{I+\eta A, x_0 + \eta b} \right\rangle - \left| \left( \frac{I+\eta A}{\|I+\eta A\|} \right)^{-1}(x_0 + \eta b) \right\rangle}_{\text{QLSP\_solver error} \leqslant \varepsilon_1 \text{ (c.f., Proposition 1)}} + \underbrace{\left| \left( \frac{I+\eta A}{\|I+\eta A\|} \right)^{-1}(x_0 + \eta b) \right\rangle - \left| A^{-1}b \right\rangle}_{\text{PPA error} \leqslant \varepsilon_2/\Psi \text{ (c.f., Proposition 2)}} \right\| \leqslant \varepsilon.$$

*Suppose the existence of a* `QLSP_solver` *satisfying the assumptions of Proposition 1 such that* (10) *is satisfied with* $\varepsilon_1 = \varepsilon/c$, *for* $c > 1$. *Further, suppose the assumptions of Proposition 2 hold, i.e., single-run PPA with* $\eta = \kappa \left( \frac{d}{\varepsilon_2} - 1 \right)$ *is implemented with accuracy* $\varepsilon_2 = \left( 1 - \frac{1}{c} \right) \varepsilon \cdot \Psi$. *Then, the output of Algorithm 1 satisfies:*

$$\left\| \left| \psi_{I+\eta A, x_0 + \eta b} \right\rangle - \left| A^{-1} b \right\rangle \right\| \leqslant \varepsilon.$$

*Moreover, the dependence on the condition number of* `QLSP_solver` *changes from* $\kappa$ *to* $\frac{\kappa(1+\eta)}{\kappa+\eta}$.

We now further interpret the analysis from the previous subsection and compare it with other QLSP algorithms (e.g., CKS (Childs et al., 2017)). We first recall the CKS query complexity below.

**Theorem 4** (CKS complexity, Theorem 5 in (Childs et al., 2017)). *The QLSP in Definition 1 can be solved to* $\varepsilon$-*accuracy by a quantum algorithm that makes* $\mathcal{O}\left( \kappa \cdot \text{poly} \log\left( \frac{\kappa}{\varepsilon} \right) \right)$ *queries to the oracles* $\mathcal{P}_A$ *in Definition 3 and* $\mathcal{P}_B$ *in Definition 2.*

It was an open question whether there exists a quantum algorithm that can match the lower bound: $\Omega\left( \kappa \cdot \log\left( \frac{1}{\varepsilon} \right) \right)$ (Orsucci & Dunjko, 2021). A recent quantum algorithm based on the complicated discrete adiabatic theorem (Costa et al., 2022) was shown to match this lower bound. We recall their result below.

**Theorem 5** (Optimal complexity, Theorem 19 in (Costa et al., 2022)). *The QLSP in Definition 1 can be solved to* $\varepsilon$-*accuracy by a quantum algorithm that makes* $\mathcal{O}\left( \kappa \cdot \log\left( \frac{1}{\varepsilon} \right) \right)$ *queries to the oracles* $\mathcal{U}_A$ *in Definition 4 and* $\mathcal{P}_B$ *in Definition 2.*

A natural direction to utilize Algorithm 1 is to use the best `QLSP_solver` (Costa et al., 2022), which has the query complexity $\mathcal{O}\left( \kappa \cdot \log\left( \frac{1}{\varepsilon} \right) \right)$ from Theorem 5. We summarize this in the next theorem.

**Theorem 6** (Improving the optimal complexity). *Consider running Algorithm 1 with (Costa et al., 2022) as the candidate for* `QLSP_solver`, *which has the original complexity of* $\mathcal{O}\left( \kappa \cdot \log\left( \frac{1}{\varepsilon} \right) \right)$ *(c.f., Theorem 5). The modified complexity of Algorithm 1 via Theorem 3 can be written and decomposed to:*

$$\hat{\kappa} \cdot \log\left( \frac{c}{\varepsilon} \right) = \underbrace{\frac{\kappa(1+\eta)}{\kappa+\eta} \cdot \log\left( \frac{1}{\varepsilon} \right)}_{\text{Improvement}} + \underbrace{\frac{\kappa(1+\eta)}{\kappa+\eta} \cdot \log(c)}_{\text{Overhead}}, \tag{17}$$

*where the "improvement" comes from* $\hat{\kappa} \leqslant \kappa$, *and the "overhead" is due to the weight of* $\varepsilon_1 = \varepsilon/c$ *(and subsequently* $\varepsilon_2 = \left( 1 - \frac{1}{c} \right) \varepsilon \cdot \Psi$) *in Theorem 3. Further, with* $\eta = \kappa\left( \frac{d}{\varepsilon_2} - 1 \right)$, *it follows*

$$\hat{\kappa} = \frac{\kappa(1+\eta)}{\kappa+\eta} = \kappa - \frac{(c-1)(\kappa-1)\Psi\varepsilon}{c \cdot d} \leqslant \kappa. \tag{18}$$

We illustrate Theorem 6 in Figure 2 (left); thanks to the flexibility of Algorithm 1 and Theorem 3, similar analysis can done with CKS (Childs et al., 2017) as the baseline, as illustrated in Figure 2 (right).

Intuitively, based on the decomposition in (17), we can see that the constant $c$ –which controls the weight of $\varepsilon_1$ and $\varepsilon_2$ in Theorem 3– enters a logarithmic term in the "overhead." On the contrary, the (additive) improvement in $\kappa$ is proportional to the term $\frac{(c-1)}{c}$, as can be seen in (18).

## 5 CONCLUSION AND DISCUSSION

In this work, we proposed a novel quantum algorithm for solving the quantum linear systems problem (QLSP), based on the proximal point algorithm (PPA). Specifically, we showed that implementing a single-step PPA is possible by utilizing existing QLSP solvers. We designed a meta-algorithm where any QLSP solver can be utilized as a subroutine to improve the dependence on the condition number. Even the current best quantum algorithm (Costa et al., 2022) can be significantly accelerated via Algorithm 1, especially when the problem is ill-conditioned.

**Limitations.** A main limitation of this work is that only constant-level improvement is attained; yet, due to the existing lower bound (Harrow et al., 2009; Orsucci & Dunjko, 2021), asymptotic improvement is not possible. Another limitation is that only a single-step PPA is implemented in Algorithm 1 at the moment. This necessitates $\eta$ to be fairly large, as can be seen in (7) or (15). Then, in conjunction with Remark 3 and (18), the benefit from the modified condition number $\hat{\kappa} = \frac{\kappa(1+\eta)}{\kappa+\eta}$ diminishes. To address this, two interesting future directions can be considered.

**Implementing multi-step PPA.** A natural idea is to implement multi-step PPA. Based on (7), two-step PPA can be written as :

$$
\begin{aligned}
x_{t+1} &= (I + \eta A)^{-2}(x_{t-1} + \eta b) + (I + \eta A)^{-1}\eta b \\
&= \big((I + \eta A)^{-2} + (I + \eta A)^{-1}\big)\eta b,
\end{aligned}
\tag{19}
$$

where in the second step we set $x_{t-1}$ with zero vector (i.e., initialization). As can be seen, this requires implementing different powers (and their addition) of the modified matrix $I + \eta A$, which is possible via Gilyén et al. (2019, Lemmas 52 and 53).

In particular, one can show that the two-step PPA in (19) can satisfy a similar guarantee to Propostion 2 with smaller step size: $\eta \geqslant \kappa\left(\sqrt{\frac{d}{\varepsilon_2}} - 1\right)$. This leads to a bigger improvement in the modified condition number:

$$
\hat{\kappa} = \frac{\kappa(1+\eta)}{\kappa+\eta} = \kappa - (\kappa-1)\sqrt{\frac{(c-1)\Psi\varepsilon}{c \cdot d}}.
\tag{20}
$$

However, we need to implement the addition of two block-encoded matrices that polynomially approximate the inverse function (c.f., (19)), which roughly doubles the query complexity; it seems unlikely that the improvement in $\hat{\kappa}$ provided by two-step PPA in (20) can compensate for the overhead of doubling the query complexity. Still, single-step PPA provides significant constant-level improvement via warm starting, as we illustrate below.

**Warm starting.** The modified condition number $\hat{\kappa}$ of $(I + \eta A)/\|I + \eta A\|$ relies on the initial point of PPA, $x_0$. In particular, $\hat{\kappa}$ can be expressed as in (18) from Theorem 6, where $d := \|x_0 - x^\star\|$. Therefore, a better initialization $x_0$ via warm starting can result in a bigger improvement in the overall complexity.

Let us provide a simple example. Suppose we initialize $x_0$ such that $d := \|x_0 - x^\star\| = 2 \cdot \frac{\kappa-1}{\kappa} \cdot \varepsilon_2$. This is possible since $\varepsilon_2$ is the level of error achieved by the (classical) PPA with the specified step size $\eta$ (c.f., Proposition 2). As $\frac{\kappa-1}{\kappa} \approx 1$ for large $\kappa$, one can simply run a few iterations of classical optimization (e.g., gradient descent), and initialize $x_0$ with the output that satisfies roughly twice the desired $\varepsilon_2$ .

Then, based on (18), we have

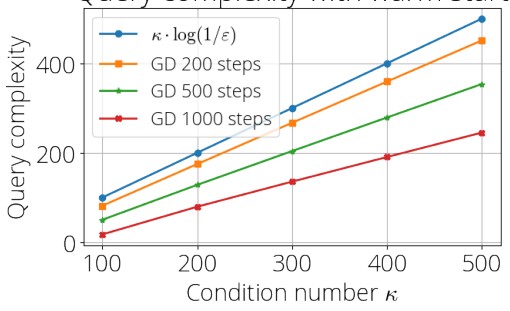

Figure 3: *Query complexity improvement with warm start.* Baseline (blue) is the optimal query complexity $\Omega(\kappa \log \frac{1}{\varepsilon})$ (Costa et al., 2022). The other three lines are the improved query complexity in (17) using Algorithm 1 initialized with $\{200, 500, 1000\}$ steps of gradient descent.

$$
\hat{\kappa} = \frac{\kappa(1+\eta)}{\kappa+\eta} = \kappa - \frac{(c-1)(\kappa-1)\Psi\varepsilon}{c \cdot d} \xrightarrow{d \leftarrow \frac{2(\kappa-1)\varepsilon_2}{\kappa}} \kappa - \frac{\kappa}{2} = \frac{\kappa}{2}.
$$

That is, simply by running a few steps of gradient descent classically such that $\|x_{\text{GD}} - x^\star\| \approx 2 \cdot \varepsilon_2$, and initializing $x_0 \leftarrow x_{\text{GD}}$ in Algorithm 1, the overall query complexity can be halved compared to the SOTA quantum algorithm (Costa et al., 2022).

We illustrate this in Figure 3. We generate (normalized) $A$ and $b$ from $\mathcal{N}(0,1)$. We vary the condition number of $A$ to be $\kappa \in \{100, 200, 300, 400, 500\}$. We plot the overall query complexity of Algorithm 1 with warm start, where $x_0$ is initialized with the last iterate of $\{200, 500, 1000\}$ steps of gradient descent. The baseline (blue) is the optimal query complexity $\Omega(\kappa \log \frac{1}{\varepsilon})$ (Costa et al., 2022), which can be effectively halved (red) via Algorithm 1.

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

# SUPPLEMENTARY MATERIALS FOR
## "A CATALYST FRAMEWORK FOR THE QUANTUM LINEAR SYSTEM PROBLEM VIA THE PROXIMAL POINT ALGORITHM"

# A  BACKGROUND ON QLSP

## A.1  DETAILS ON HHL AND CKS ALGORITHMS

Before we delve into the details of the proposed method, we briefly review the HHL and the CKS algorithms. In short, such previous approaches deal with *unitarily* mapping $x \mapsto \frac{1}{x}$.

**An overview of the HHL algorithm.** Recalling the goal of this task, we are looking for the solution vector $x^\star = A^{-1}b$. For simplicity, we assume $A$ is Hermitian with its eigenvalues in the range $[1/\kappa, 1]$, i.e., $A$ is positive definite, with condition number $\kappa$. As $A$ is Hermitian, it admits the spectral decomposition:

$$A = \sum_{j=1}^{N} \lambda_j u_j u_j^*,$$

where $\{u_j\}_{j=1}^{N}$ are the (orthonormal) eigenvector bases, and $\{\lambda_j\}_{j=1}^{N}$ the corresponding eigenvalues. Then, $A^{-1}$ can be written with respect to these same bases as

$$A^{-1} = \sum_{j=1}^{N} \frac{1}{\lambda_j} u_j u_j^*.$$

Similarly, since $\{u_j\}_{j=1}^{N}$ form an orthonormal basis, the vector $b$ can be written as:

$$b = \sum_{j=1}^{N} \beta_j u_j.$$

Putting together the above, the solution vector $x^\star = A^{-1}b$ is written as:

$$A^{-1}b = \sum_{j=1}^{N} \beta_j \frac{1}{\lambda_j} u_j. \tag{21}$$

However, the maps $A$ and $A^{-1}$ are generally not unitary, preventing us from directly applying these maps "quantumly" to the state $|b\rangle$. To that end, one of the main observations of HHL was that $U = e^{iA}$ (where $i$ stands for the imaginary unit) is indeed unitary and has the same eigenvectors as $A$ (and $A^{-1}$). Therefore, utilizing the "hamiltonian simulation" to implement $U = e^{iA}$, and "phase estimation" to estimate $\lambda_j$ associated with the eigenvector $|a_j\rangle$, one can unitarily map:

$$\sum_j \beta_j |a_j\rangle \left( \frac{1}{\kappa \lambda_j} |0\rangle + \sqrt{1 - \frac{1}{(\kappa \lambda_j)^2}} |1\rangle \right) \quad \longmapsto \quad \underbrace{\frac{1}{\kappa} \sum_j \beta_j \frac{1}{\lambda_j} |a_j\rangle |0\rangle}_{\text{Proportional to } A^{-1}b} + |\phi\rangle |1\rangle.$$

Finally, applying $O(\kappa)$ rounds of amplitude amplification, we can amplify this part of the state to 1. Some of these subroutines are reviewed in the next subsection.

**An overview of the CKS contributions.** Yet, the phase estimation is still a bottleneck. To address this, Childs-Kothari-Somma (CKS hereafter) (Childs et al., 2017) improved the complexity of HHL, where a significant improvement in the suboptimality is due to the Linear Combination of Unitaries (LCU) and the variable time amplitude amplification from (Ambainis, 2012). We recall the (general) LCU lemma below:

**Lemma 7** (Lemma 7 in (Childs et al., 2017)). *Let $M = \sum_i \alpha_i T_i$ with $\alpha_i > 0$ for some operators $\{T_i\}$ which can be not necessarily unitary. Let $\{U_i\}$ be a set of unitaries such that $U_i|0^t\rangle|\phi\rangle = |0^t\rangle T_i|\phi\rangle + |\Phi_i^\perp\rangle$ for all states $|\phi\rangle$, where $t > 0$ is an integer and $(|0^t\rangle\langle 0^t| \otimes \mathbb{I})|\Phi_i^\perp\rangle = 0$. Given a procedure $\mathcal{P}_b$ for creating a state $|b\rangle$, there exists a quantum algorithm that exactly prepares the quantum state $M|b\rangle/\|M|b\rangle\|$ with constant success probability making in expectation $O(\alpha/\|M|b\rangle\|)$ uses of $\mathcal{P}_b$, $U := \sum_i |i\rangle\langle i| \otimes U_i$, and $V$ such that $V|0^m\rangle = \frac{1}{\sqrt{\alpha}} \sum_i \sqrt{\alpha_i}|i\rangle$.*

In particular, CKS utilized (truncated) Chebyshev polynomials to choose $T_i$'s in $M = \sum_i \alpha_i T_i$, to approximate $1/x$ unitarily. The polynomial they used is the degree-$(t-1)$ Taylor expansion of $1/x$ around the point $x = 1$:

$$p_t^{\text{CKS}} := \sum_{k=0}^{t-1} (1-x)^k = \frac{1-(1-x)^t}{x}. \tag{22}$$

The approximation of the above polynomial is characterized in the Lemma below:

**Lemma 8** ((Childs et al., 2017)). *For all $x \in [1/\kappa, 1]$, $p_t^{CKS}$ with $t \geqslant \kappa \log(\kappa/\varepsilon)$ satisfies $|p_t^{CKS} - \frac{1}{x}| \leqslant \varepsilon$.*

Gribling-Kerenidis-Szilágyi (Gribling et al., 2021) further improved the previous result by utilizing the *optimal* polynomial, which are scaled Chebyshev polynomials (Sachdeva et al., 2014), in similar spirit to the Chevyshev iterative method in the classical optimization.

### A.2 SOME QUANTUM SUBROUTINES USED IN HHL (HARROW ET AL., 2009) AND CKS (CHILDS ET AL., 2017)

---

**Algorithm 2** Hamiltonian Simulation

---

**Require:** Hermitian matrix $H$, evolution time $t$, error tolerance $\epsilon$, number of terms $m$
**Ensure:** Approximation of $e^{-iHt}$
 1: Decompose $H$ into $m$ terms: $H = \sum_{j=1}^m H_j$, where each $H_j$ is easy to simulate
 2: Choose $r$, the number of Trotter-Suzuki steps, such that error $\leqslant \epsilon$
 3: **for** $k = 1, 2, \ldots, r$ **do**
 4: $\quad U_k \leftarrow e^{-iH_1 t/r} e^{-iH_2 t/r} \cdots e^{-iH_m t/r}$ $\qquad\qquad$ ▷ First-order Trotter-Suzuki decomposition
 5: **end for**
 6: Combine all $U_k$ to form $U = \prod_{k=1}^r U_k$
 7: **return** $U$ as the approximate evolution operator

---

**Algorithm 3** Phase Estimation

---

**Require:** $U$: Unitary operator
**Require:** $|\psi\rangle$: Eigenvector of $U$
**Require:** $t$: Number of qubits for precision
 1: Initialize $t$ qubits in state $|0\rangle^{\otimes t}$
 2: Apply Hadamard gate to each qubit
 3: Apply controlled-$U^{2^j}$ gate for each qubit, where $j$ is the qubit index
 4: Apply inverse Quantum Fourier Transform (QFT)
 5: Measure the $t$ qubits
 6: Let $m$ be the measurement outcome
 7: **return** $m/2^t$ $\qquad\qquad\qquad\qquad\qquad\qquad\qquad\qquad\qquad\qquad\qquad$ ▷ Estimated phase

---

---

**Algorithm 4** Amplitude Amplification

---

**Require:** $|\psi\rangle$: Initial state
**Require:** $\mathcal{O}$: Oracle operator
**Require:** $T$: Number of iterations
1: $|s\rangle \leftarrow \frac{1}{\sqrt{N}}\sum_x |x\rangle$          ▷ Initialize uniform superposition
2: $|\psi'\rangle \leftarrow |s\rangle$
3: **for** $t = 1$ to $T$ **do**
4:      $|\psi'\rangle \leftarrow 2|s\rangle\langle s| - I|\psi'\rangle$          ▷ Reflection about $|s\rangle$
5:      $|\psi'\rangle \leftarrow \mathcal{O}(|\psi'\rangle)$          ▷ Apply the oracle operator
6:      $|\psi'\rangle \leftarrow 2|s\rangle\langle s| - I|\psi'\rangle$          ▷ Reflection about $|s\rangle$
7: **end for**
8: **return** $|\psi'\rangle$          ▷ Amplified state

---

**Algorithm 5** Linear Combination of Unitaries

---

**Require:** $U_1, U_2, \ldots, U_m$: unitary operators, $\alpha_1, \alpha_2, \ldots, \alpha_m$: complex coefficients
**Ensure:** $U = \sum_{j=1}^m \alpha_j U_j$
1: Prepare an $(m + 1)$-qubit state $|0\rangle|\psi_0\rangle$, where $|\psi_0\rangle = \sum_{j=1}^m \sqrt{p_j}|j\rangle$ with $p_j = |\alpha_j|^2 / \sum_{k=1}^m |\alpha_k|^2$
2: **for** $j = 1 \ldots m$ **do**
3:      Apply the controlled unitary operation $C^{(j)}(U_j)$, where $C^{(j)}$ is controlled by the $j$th qubit of the first register
4: **end for**
5: Perform the inverse quantum Fourier transform on the first register
6: Measure the first register to obtain an integer $0 \leqslant k \leqslant 2^m - 1$
7: Compute $U_k = \sum_{j=1}^m \alpha_j U_j^k \pmod{2^m}$
8: Apply $U_k$ to the second register
9: **return** the state of the second register

---

# B MISSING PROOFS

## B.1 PROOF OF LEMMA 1

*Proof.* By the assumption on QLSP in Definition 1, the singular values of $A$ is contained in the interval $\left[\frac{1}{\kappa}, 1\right]$. Thus, for the normalized modified matrix, i.e., $\frac{I+\eta A}{\|I+\eta A\|}$, the singular values are contained in the interval $\left[\frac{\kappa+\eta}{\kappa(1+\eta)}, 1\right]$, by spectral mapping theorem. This leads to the modified condition number $\hat{\kappa} = \frac{\kappa(1+\eta)}{\kappa+\eta}$. □

## B.2 PROOF OF LEMMA 2

*Proof.* We need to bound:

$$\|\hat{x} - \hat{y}\| = \left\| \frac{x}{\|x\|} - \frac{y}{\|y\|} \right\|.$$

Consider the inner product form:

$$\left\| \frac{x}{\|x\|} - \frac{y}{\|y\|} \right\|^2 = 2 - 2\left\langle \frac{x}{\|x\|}, \frac{y}{\|y\|} \right\rangle = 2 - 2\frac{\langle x, y \rangle}{\|x\|\|y\|}.$$

The inner product can be bounded using $\|x - y\| \leqslant \epsilon$:

$$\langle x, y \rangle = \frac{1}{2}\left( \|x\|^2 + \|y\|^2 - \|x - y\|^2 \right)$$

$$\geqslant \frac{1}{2}\left( \|x\|^2 + \|y\|^2 - \epsilon^2 \right).$$

Now, substituting this into the normalized inner product:

$$\frac{\langle x, y \rangle}{\|x\|\|y\|} \geqslant \frac{\frac{1}{2}\left(\|x\|^2 + \|y\|^2 - \epsilon^2\right)}{\|x\|\|y\|} = \frac{1}{2}\left(\frac{\|x\|}{\|y\|} + \frac{\|y\|}{\|x\|} - \frac{\epsilon^2}{\|x\|\|y\|}\right).$$

Using $\frac{\|x\|}{\|y\|} + \frac{\|y\|}{\|x\|} \geqslant 2$ from AM-GM inequality, we get:

$$\frac{\langle x, y \rangle}{\|x\|\|y\|} \geqslant 1 - \frac{\epsilon^2}{2\|x\|\|y\|}.$$

Substituting the above back into the distance expression, we get:

$$\left\|\frac{x}{\|x\|} - \frac{y}{\|y\|}\right\|^2 \leqslant 2 - 2\left(1 - \frac{\epsilon^2}{2\|x\|\|y\|}\right) = \frac{\epsilon^2}{\|x\|\|y\|},$$

which implies the desired result:

$$\left\|\frac{x}{\|x\|} - \frac{y}{\|y\|}\right\| \leqslant \frac{\epsilon}{\sqrt{\|x\|\|y\|}}.$$

$\square$

### B.3 Proof of Proposition 1

*Proof.* Since the input matrix $\frac{I+\eta A}{\|I+\eta A\|}$ is normalized, and due to the reasoning in Remark 1, we can have sparse-acess or block-encoding of $\frac{I+\eta A}{\|I+\eta A\|}$. Then, the result of Proposition 1 simply follows from the result of each related work in Table 1. $\square$

### B.4 Proof of Proposition 2

*Proof.* As explained in Section 4.2 of the main text, we can compute the number of iterations for PPA using (15), recalled below, which is based on (14).

$$\frac{d}{(1 + \eta/\kappa)^t} \leqslant \varepsilon_2 \implies \frac{\log(\frac{d}{\varepsilon_2})}{\log(1 + \frac{\eta}{\kappa})} \leqslant t.$$

In this work, since we are implementing a single-step PPA, we want $t = 1$ from (15). To achieve that, we can have:

$$\log\left(\frac{d}{\varepsilon_2}\right) = \log\left(1 + \frac{\eta}{\kappa}\right) \Rightarrow$$

$$\eta = \kappa\left(\frac{d}{\varepsilon_2} - 1\right).$$

It remains to invoke Lemma 2 to complete the proof. $\square$

### B.5 Proof of Theorem 3

*Proof.* Recall the decomposition:

$$\left\|\underbrace{\left|\psi_{I+\eta A, x_0 + \eta b}\right\rangle - \left|\left(\frac{I+\eta A}{\|I+\eta A\|}\right)^{-1}(x_0 + \eta b)\right\rangle}_{\texttt{QLSP\_solver error} \leqslant \varepsilon_1} + \underbrace{\left|\left(\frac{I+\eta A}{\|I+\eta A\|}\right)^{-1}(x_0 + \eta b)\right\rangle - \left|A^{-1}b\right\rangle}_{\text{PPA error} \leqslant \varepsilon_2}\right\| \leqslant \varepsilon.$$

The $\texttt{QLSP\_solver}$ error can be provided by Proposition 1, with the accuracy of $\varepsilon_1 = \varepsilon/c$, for $c > 1$. Similarly, $\eta$ is chosen based on Proposition 2, with $\varepsilon_2 = \left(1 - \frac{1}{c}\right)\varepsilon \cdot \Psi$. Then, by adding the result from each proposition, we have

$$\varepsilon_1 + \frac{\varepsilon_2}{\Psi} = \frac{\varepsilon}{c} + \left(1 - \frac{1}{c}\right)\varepsilon = \varepsilon.$$

Note that since $\texttt{QLSP\_solver}$ is only called once in Algorithm 1, the final accuracy $\varepsilon$ is the same for the output of Algorithm 1 and the subroutine $\texttt{QLSP\_solver}$ being called. $\square$

### B.6 PROOF OF THEOREM 6

(17) simply follows from plugging in the modified condition number $\hat{\kappa}$, along with $\varepsilon_1 = \varepsilon/c$. That is,

$$\mathcal{O}\left(\hat{\kappa} \cdot \log\left(\frac{1}{\varepsilon_1}\right)\right) = \mathcal{O}\left(\frac{\kappa(1+\eta)}{\kappa+\eta} \cdot \log\left(\frac{c}{\varepsilon}\right)\right)$$

$$= \mathcal{O}\left(\frac{\kappa(1+\eta)}{\kappa+\eta} \cdot \log\left(\frac{1}{\varepsilon}\right) + \frac{\kappa(1+\eta)}{\kappa+\eta} \cdot \log(c)\right).$$

Note that similar analysis can be done for CKS (Childs et al., 2017), starting from $\mathcal{O}\left(\hat{\kappa} \cdot \log\left(\frac{\hat{\kappa}}{\varepsilon_1}\right)\right)$.

For (18), it follows from the below steps:

$$\frac{\kappa(1+\eta)}{\kappa+\eta} = \frac{1+\eta}{1+\eta/\kappa}$$

$$= \frac{1 + \frac{\kappa dc}{(c-1)\varepsilon\Psi} - \kappa}{\frac{\kappa dc}{(c-1)\varepsilon\Psi}}$$

$$= \kappa - \frac{\kappa-1}{\frac{dc}{(c-1)\varepsilon\Psi}}$$

$$= \kappa - \frac{(c-1)\varepsilon\Psi(\kappa-1)}{dc}.$$

