# OpenReview forum: "A Catalyst Framework for the Quantum Linear System Problem via the Proximal Point Algorithm"
_ICLR.cc/2025/Conference — Submitted to ICLR 2025_

### Official Review · Reviewer_eGnX · 2024-10-23

**Soundness:** 3
**Presentation:** 2
**Contribution:** 3
**Rating:** 6
**Confidence:** 3

**Summary:**

Quantum linear system problem (QLSP) is of great importance in quantum algorithms, which focuses on solving a system of linear equations, i.e., find a state $\ket{x} = x/\|x\|$ where $x$ is the solution to $Ax=b$. QLSP problem is fundamental as it is a key step in many quantum algorithms, such as quantum recommendation systems and solving differential equations. However, a major limitation of the QLSP solver is that the complexity has a linear dependence of the condition number $\kappa$ of the matrix $A$. Whether one can alleviate this issue is also of general interest.

In this paper, the authors propose a new algorithm based on proximal point algorithm (PPA) to reduce the query complexity in solving QLSP in a constant degree. To achieve this acceleration, the authors uses the technique of shifting the matrix $A$ of the problem to $I+\eta A$, leading to the change of condition number of the problem. In addition, the authors also draw some graphs to demonstrate the acceleration.

**Strengths:**

The authors propose a new algorithm (PPA for QLSP) to alleviate the linear dependence on condition number of the QLSP solvers, yielding a constant acceleration of previous QLSP solvers.

**Weaknesses:**

First, a main technical tool is the use of PPA algorithm for shifting, which might lack novelity.
Second, the proofs of the main theorems seems straigthforward and incremental.
Also, it looks like more numerical experiments could be conducted to demonstrate the actual performance of their proposed algorithm.
Furthermore, I am not sure whether the topic of this paper fits for the scope of ``learning represenations''.

**Questions:**

- In Theorem 3, the parameter $\eta$ is set to $\kappa (d/\varepsilon_2 - 1)$, but $\kappa$ may be unknown. Could the authors explain what to to if $\kappa$ is not known?

- In Theorem 6, it seems that difference between the modified condition number $\hat{\kappa}$ and $\kappa$ has a linear dependence on $\Psi$, which relies on the choice of $x_0$ and $\eta$. Could the authors explain more about how to choose $x_0$ and $\eta$?

- Page 9, Line 482: proximal poin algorithm —> proximal point algorithm.

- Page 14, Line 733: the codes in algorithm 2 for Hamilonian simulation could be improved for readability.

---

> ### Author Response · Authors · 2024-11-23
>
> We are pleased to hear that you found our approach novel.
>
> Thank you for your comments and feedback. We address your comments and questions below in detail.
>
> > First, a main technical tool is the use of PPA algorithm for shifting, which might lack novelity.
>
> While it's true that the proximal point algorithm (PPA) is a well-studied method in classical optimization, its application to quantum linear systems problem (QLSP) is entirely new. In fact, applying any iterative procedure to solving QLSP is new; all previous works focus on how to estimate $A^{-1}$ efficiently, and hence there is no way to tune or improve the practical performance, given a fixed $A$. As a result, it provides a new perspective on tackling the condition number problem in QLSP.
>
> We provide some examples of highly impactful papers that combine known techniques from distant fields:
> - The optimal quantum algorithm [1] which matches the lower bound $O(\kappa \log(1/\varepsilon))$ is achieved by applying the “adiabatic quantum computation” for the “QLSP problem”
> - A highly influential paper in deep learning [2] applies the “heavyball momentum,” a well-known technique in optimization, in “training neural networks.” Yet, this work has initiated the popularity of using momentum for training deep neural networks, which has now become the norm.
> - Alphafold [3] applies “deep neural networks,” which is known, to the “protein folding problem.” Yet, it was highly influential as it was done for the first time.
>
> Similarly, our contribution lies in the novel integration of this classical iterative method with quantum algorithms for the fundamental problem, creating a hybrid approach that enables users to tune the parameter $\eta$ to improve the practical performance of any existing QLSP solver.
>
>
> > Second, the proofs of the main theorems seems straigthforward and incremental.
>
> Thank you for the comment. We actually view this simplicity as one of the main strengths of our work: by simply "wrapping" our meta-algorithm with a given QLSP solver, one can achieve **a significant constant level improvement**. The simplicity in proof reflects the simplicity of our method.
>
> > Also, it looks like more numerical experiments could be conducted to demonstrate the actual performance of their proposed algorithm.
>
> Unfortunately, we do not have access to an actual quantum computer, so we cannot provide numerical experiments of the actual performance of the proposed algorithm. That is why we tried to provide numerical illustration of the quantities from theory, such as Figure 1 and Figure 2.
>
> > Furthermore, I am not sure whether the topic of this paper fits for the scope of ``learning represenations''.
>
> Our work is particularly relevant to ICLR and the machine-learning community for several reasons: (quantum) linear systems are fundamental to many (quantum) machine learning algorithms, including (quantum) support vector machines, (quantum) principal component analysis, and (quantum) recommendation systems.
>
> Our meta-algorithm approach introduces a new framework for improving quantum algorithms, which could inspire similar hybrid classical-quantum approaches in other areas of quantum machine learning. The trade-off between runtime and approximation error that our method introduces is a common theme in machine learning, where balancing computational efficiency and model accuracy is crucial.
>
>
> > In Theorem 3, the parameter $\eta$
>  is set to $\kappa(d/\varepsilon_2 - 1)$, but $\kappa$ may be unknown. Could the authors explain what to to if $\kappa$ is not known?
>
> We first note that knowledge of the condition number of $A$, is required by all previous quantum algorithms [1, 4, 5].
>
> That being said, we would like to remind about the "interpolation" detailed in Remark 3. The step size $\eta$ continuously interpolates the original problem (for $\eta = + \infty$) to a pre-conditioned problem (for small $\eta$). That is, in practice, one can use "sufficiently large" $\eta$, and can only benefit from our meta algorithm. That is, in practice, one can use "sufficiently large" $\eta$, and can only benefit from our meta algorithm.
>
> > In Theorem 6, it seems that difference between the modified condition number $\hat{\kappa}$ and $\kappa$ has a linear dependence on $\Psi$, which relies on the choice of $x_0$ and $\eta$. Could the authors explain more about how to choose $x_0$ and $\eta$ ?
>
> $\eta$ can be set to be sufficiently large, as in the previous comment. For $x_0$, in pg. 10 of our updated manuscript (teal color), we show how to halve the overall complexity of the current SOTA quantum algorithm [1] simply by using "warm up" (i.e., smarter initialization).
>
>
> > Page 9, Line 482: proximal poin algorithm —> proximal point algorithm.
>
> Thank you for pointing out. We have fixed the typo.
>
> > Page 14, Line 733: the codes in algorithm 2 for Hamilonian simulation could be improved for readability.
>
> Thank you for the comment. We have modified the pseudocode to improve readability.

---

> > ### Author Response · Authors · 2024-11-23
> >
> > We hope this clarifies your concerns. Please don't hesitate to follow up with unresolved concerns.
> >
> > [1] P. Costa et al. “Optimal scaling quantum linear-systems solver via discrete adiabatic theorem”
> >
> > [2] I. Sutskever, J. Martens, G. Dahl, and G. Hinton. “On the importance of initialization and momentum in deep learning.”
> >
> > [3] J. Jumper et al. “Highly accurate protein structure prediction with AlphaFold”
> >
> > [4] P. Costa et al. “Optimal scaling quantum linear-systems solver via discrete adiabatic theorem”
> >
> > [5] A. Harrow et al. “Quantum algorithm for linear systems of equations”

---

> ### Comment · Reviewer_eGnX · 2024-11-26
>
> Thank you for your comprehensive response. Your clarifications have successfully addressed my previous concerns, and I have adjusted my scores accordingly.
>
> However, concerning recent developments on quantum linear system problem solving [1], I would appreciate further elaboration on the computational complexity aspects of your algorithm, specifically:
>
> - The query complexity associated with the block encoding oracle of A
> - The query complexity related to the state preparation oracle of b
>
> A detailed breakdown of these individual complexity components would help in better understanding the overall efficiency of your approach.
>
> [1] Low, G. H., & Su, Y. (2024). Quantum linear system algorithm with optimal queries to initial state preparation. arXiv preprint arXiv:2410.18178.

---

> > ### Author Response · Authors · 2024-11-27
> >
> > Thank you for the follow up question. We remind the reviewer that **our proposed method is a meta-algorithm agnostic of different QLSP_solvers**. Thus, one can plug in [1] by Low and Su (2024) as the QLSP_solver within our proposed algorithm, and achieve significant constant level improvement in $\kappa$, **while maintaining essentially the same query complexity (to $O_A$ and $O_b$) as the original QLSP_solver.**
> >
> > Specifically, in [1], Low and Su (2024) provide another QLSP_solver where the query complexity to $O_b$ (state preparation oracle) is smaller than the query complexity to $O_A$, which is $\Theta(\kappa\log (\frac{1}{\varepsilon}))$ (c.f., Table 1 in Low and Su (2024)). We note that the overall query complexity is still dominated by $\Theta(\kappa\log \left(\frac{1}{\varepsilon}\right))$, which is the same as [2].
> >
> > That being said, one can simply plug in [1] in line 2 of Algorithm 1 in our manuscript. Combined with our meta-algorithm, one can benefit *both* from the reduced query complexity to $O_b$ thanks to [1], and improved dependence on the condition number from $\kappa$ to $\hat{\kappa}$ via our Algorithm 1, similarly to Theorem 6 in our manuscript.
> >
> >
> > We also want to point you to the relevant numerical simulation update we made in the manuscript. **In pg 10 (Figure 3)**, we included simulation results showcasing that **the optimal query complexity $\Theta(\kappa \log \frac{1}{\varepsilon})$ can effectively be halved via warm starting with gradient descent.**
> >
> > Specifically, we generate (normalized) $A$ and $b$ from $\mathcal{N}(0, 1)$, and vary the condition number of $A$ to be $\kappa \in \\{100, 200, 300, 400, 500\\}$. We plot the overall query complexity of our proposed method with warm start, where $x_0$ is initialized with the last iterate of $\\{200, 500, 1000\\}$ steps of gradient descent. One can see the overall query complexity gradually improves with better initialization, and can effectively halve the overall cost with 1000 steps of gradient descent. **We remind that "warm starting" or "better initialization" is NOT possible with any other existing QLSP solver.**
> >
> > We hope this addition clarifies your concern, and if so, we hope you could consider reflecting on the score. Thank you again for your time in reviewing our manuscript.
> >
> >
> > [1] Low, G. H., & Su, Y. (2024). "Quantum linear system algorithm with optimal queries to initial state preparation."
> >
> > [2] P. Costa et al. (2022) “Optimal scaling quantum linear-systems solver via discrete adiabatic theorem”

---

> > > ### Author Response · Authors · 2024-12-02
> > >
> > > Dear Reviewer eGnX,
> > >
> > > Thank you again for your constructive feedback and thoughtful review. Given that the discussion period is ending soon, we would greatly appreciate if you could check out our **response to your concern as well as the warm-starting result that we summarized in the above comment.**
> > >
> > > We hope this clarifies your concern, and if so, we hope you could consider reflecting on the score. Thank you again for your time in reviewing our manuscript.

---

### Official Review · Reviewer_yCYL · 2024-11-03

**Soundness:** 2
**Presentation:** 3
**Contribution:** 2
**Rating:** 5
**Confidence:** 4

**Summary:**

This paper is a good attempt to make QLSA more practical although there is no theoretical improvement in the bigO complexity. However, the authors did not discuss how large the “constant” improvement could be if someone wants to “read out” the quantum solution. In fact, an issue would appear in that regime and limit the “constant” improvement.

**Strengths:**

The proposed approach is easy to plug in any quantum linear systems solver. Improving the conditional number is one of the most important tasks in QLSA.

**Weaknesses:**

As demonstrated in Figure 1, kappa^hat is less than kappa/2. According to lemma 1, parameter eta would be less than kappa-2 (or simply kappa). According to the eta choice in theorem 3, if we simply use eta less than kappa, then we have epsilon_2 larger than d. Recall that epsilon_2 is the accuracy for x_{t+1}-x^* as in equation 14, which is the accuracy people care about in the classical setting. This means, if someone wants to eventually read out the quantum solution, ignoring the error from the reading-out process, the accuracy of the classical solution will be very low because the accuracy is at most d and d is simply the initial accuracy. If we don’t want to read out the solutions, the applicability will be significantly limited.

A side comment: the notations of a quantum state are not consistent, i.e., some of them use the braket notation while some do not.

**Questions:**

See the weakness above.

---

> ### Author Response · Authors · 2024-11-23
>
> Thank you for your comments  feedback. We are pleased to hear that you found our approach addresses one of the most imprtant tasks in QLSA: improving the $\kappa$ dependence.
>
> We address your comments and questions below in detail.
>
> > As demonstrated in Figure 1, $\hat{\kappa}$ is less than $\kappa/2$. According to lemma 1, parameter $\eta$ would be less than $\kappa-2$ (or simply $\kappa$). According to the $\eta$ choice in theorem 3, if we simply use $\eta$ less than $\kappa$, then we have $\epsilon_2$ larger than $d$. Recall that $\epsilon_2$ is the accuracy for $x_{t+1}-x^*$ as in equation 14, which is the accuracy people care about in the classical setting.
> >
> > This means, if someone wants to eventually read out the quantum solution, ignoring the error from the reading-out process, the accuracy of the classical solution will be very low because the accuracy is at most $d$ and $d$ is simply the initial accuracy. If we don’t want to read out the solutions, the applicability will be significantly limited.
>
> Thank you for the detailed comment. Firstly, we are unsure how you interpreted Lemma 1 as imposing $\eta$ to be less than $\kappa - 2$; Lemma 1 is independent of Algorithm 1. That being said, in Proposition 2 or Theroem 3, we set $\eta = \kappa \left( \frac{d}{\varepsilon_2} - 1 \right)$, which is larger than $\kappa$.
>
> We would like to additionally point out that the "reading out the quantum solution," in other words measuring the quantum state, is inherent to **any** quantum algorithm.
>
> On the other hand, we emphasize again that the advantage of quantum algorithm, compared to any classical algorthm, is that quantum algorithms operate with (poly)**logarithmic dependence on the dimension, i.e., $\text{poly}\log(N)$**. This is generally not possible using classical algorithm.
>
> Lastly, we note that the quantum solution for the linear system of equations can be used as a subroutine for bigger algorithms, as **Reviewer eGnX** pointed out, such as solving quantum recommendation systems [1] or solving differential equations [2], quantum SVM [3], and unsupervised learning [4].
>
>
> > A side comment: the notations of a quantum state are not consistent, i.e., some of them use the braket notation while some do not.
>
> Thank you for this comment. We will revise our manuscript to keep the notations of a quantum state consistent.
>
>
> We hope this clarifies your concerns. Please don't hesitate to follow up with unresolved concerns.
>
>
> [1] I. Kerenidis and A. Prakash (2016) "Quantum recommendation systems"
>
> [2] J.P. Liu, et al (2021) " Efficient quantum algorithm for dissipative nonlinear differential equations"
>
> [3] P. Rebentrost et al (2014) "Quantum support vector machine for big data classification"
>
> [4] N. Wiebe et al (2014) "Quantum algorithms for nearest-neighbor methods for supervised and unsupervised learning"

---

> > ### Author Response · Authors · 2024-11-27
> >
> > Dear Reviewer yCYL,
> >
> > We want to point you to another update we made in the manuscript.
> > **In pg 10 (Figure 3)**, we included simulation results showcasing that **the optimal query complexity $\Omega(\kappa \log \frac{1}{\varepsilon})$ can effectively be halved via warm starting with gradient descent.**
> >
> > Specifically, we generate (normalized) $A$ and $b$ from $\mathcal{N}(0, 1)$, and vary the condition number of $A$ to be $\kappa \in \\{100, 200, 300, 400, 500\\}$. We plot the overall query complexity of our proposed method with warm start, where $x_0$ is initialized with the last iterate of $\\{200, 500, 1000\\}$ steps of gradient descent. One can see the overall query complexity gradually improves with better initialization, and can effectively halve the overall cost with 1000 steps of gradient descent. **We remind that "warm starting" or "better initialization" is NOT possible with any other existing QLSP solver.**
> >
> > We hope this addition clarifies your concern, and if so, we hope you could consider reflecting on the score. Thank you again for your time in reviewing our manuscript.

---

> > > ### Author Response · Authors · 2024-12-02
> > >
> > > Dear Reviewer yCYL,
> > >
> > > Thank you again for your constructive feedback and thoughtful review. Given that the discussion period is ending soon, we would greatly appreciate if you could check our response to your concerns above.
> > >
> > > We hope our reply clarifies your concern, and if so, we hope you could consider reflecting on the score. Thank you again for your time in reviewing our manuscript.

---

> ### Comment · Area_Chair_sdc9 · 2024-11-27
>
> Dear Reviewer,
>
> The authors have provided their rebuttal to your comments/questions. Given that we are not far from the end of author-reviewer discussions, it will be very helpful if you can take a look at their rebuttal and provide any further comments. Even if you do not have further comments, please also confirm that you have read the rebuttal. Thanks!
>
> Best wishes,
> AC

---

### Official Review · Reviewer_w21C · 2024-11-03

**Soundness:** 3
**Presentation:** 3
**Contribution:** 2
**Rating:** 5
**Confidence:** 4

**Summary:**

This paper proposes a novel meta-algorithm for solving the quantum linear system problem (QLSP) based on the proximal point algorithm (PPA). The performance of quantum linear system solvers heavily depends on the condition number of the linear system. Due to a query lower bound result, the linear dependence on condition number can not be further improved (in the asymptotic scaling sense). This paper incorporates the proximal point algorithm as a pre-conditioner of a QLSP solver. The new proximal-point QLSS can leverage the trade-offs between the approximation error (in the output solution) and the condition number of the linear system. It is proven that a finite stepsize mitigates the condition number, at the cost of making the resulting solution inexact. Numerical results show this new approach can lead to a generic (constant) speedup in solving linear systems over existing STOA.

**Strengths:**

- This paper exploits the proximal point algorithm to reformulate the original linear system problem as an approximate optimization problem, where the new problem is parametrized by the "step size" $\eta$. This new parameter provides a continuous interpolation from the original problem (for $\eta = +\infty$) to a pre-conditioned problem (for small $\eta$). This is an interesting point of view and also the first proposal to combine the proximal point algorithm with a quantum linear system solver (to my best knowledge).
- Numerical results show that this approach reduces the total query complexity over the STOA quantum linear system solver.

**Weaknesses:**

- My main concern is that the proximal point algorithm proposed in this paper can have a **single** iteration. This feature makes this algorithm less useful in practice. The main difficulty is that the state preparation oracle (see Definition 2) is hard (or maybe impossible) to construct for subsequent steps.
- While it is discussed that a multi-step PPA can be realized by implementing different powers of the modified matrix, it is not clear why this would lead to asymptotic speedup because inverting $(I + \eta A)^n$ for $n \ge 2$ is likely to incur a super-linear overhead in the condition number. I would conjecture that the naive multi-step PPA will lead to a polynomial slowdown compared to the quantum STOA.
- I do not understand how this PPA approach is compared to an "exact" quantum linear system solver, because this PPA approach can only perform a single iteration step so the solution is not exact. Can the authors elaborate on how a fair comparison is achieved? Is the error budget for both methods pre-fixed?

**Questions:**

Besides the questions that are raised in the "Weaknesses" part, I have the following questions:
- Is there a practical way to choose the stepsize $\eta$ in the algorithm? What if the obtained solution is still far from the exact solution to the linear system?
- Is it possible to use the PPA solution as a "warm start" in a quantum linear system solver, which might be helpful to improve the overall performance?
- Can the authors further elaborate on how the comparison is made in Figure 1? Is it fair to include a $c$ parameter in Algorithm 1?

---

> ### Author Response · Authors · 2024-11-23
>
> Thank you for your comments and feedback. We are pleased to hear that you found our approach interesting, which enables a continuous interpolation based on the "step size" $\eta$.
>
> We address your comments and questions below in detail.
>
> > My main concern is that the proximal point algorithm proposed in this paper can have a single iteration. This feature makes this algorithm less useful in practice. The main difficulty is that the state preparation oracle (see Definition 2) is hard (or maybe impossible) to construct for subsequent steps.
>
> Thank you for the question. We first want to note that, even with the single-iteration proximal point algorithm, it provides two avenues to tackle the $\kappa$ bottleneck in QLSP: (i) by choosing different $\eta$, and (ii) by using "warm-start" and initialize $x_0$ more smartly. *Both of these are NOT possible with any previous quantum algorithm.* In pg. 10 of our updated manuscript, we show how to halve the overall complexity of the current SOTA quantum algorithm [1] simply by using "warm up" or smarter initialization.
>
>
> > While it is discussed that a multi-step PPA can be realized by implementing different powers of the modified matrix, it is not clear why this would lead to asymptotic speedup because inverting $(I + \eta A)^n$ for $n \geq 2$ is likely to incur a super-linear overhead in the condition number. I would conjecture that the naive multi-step PPA will lead to a polynomial slowdown compared to the quantum STOA.
>
> Continuing from the above comment, implementing multi-step proximal point algorithm indeed seems nontrivial.
>
> Specifically, we need to implement the addition of two block-encoded matrices that polynomially approximates matrix inversion (c.f., Eq. (19)), which roughly doubles the query complexity; it seems unlikely that the improvement in $\kappa$ provided by two-step PPA in Eq. (19) can compensate for the overhead of doubling the query complexity, albeit two-step PPA allows smaller $\eta$. We have summarized this in pg 10 of the updated manuscript (teal color).
>
> > I do not understand how this PPA approach is compared to an "exact" quantum linear system solver, because this PPA approach can only perform a single iteration step so the solution is not exact. Can the authors elaborate on how a fair comparison is achieved? Is the error budget for both methods pre-fixed?
>
> Thank you for the question. We are a bit confused what you mean by an "exact" quantum linear system solver. To the best of our knowledge, all the previous quantum algorithms are "approximate" (hence the $\varepsilon$ dependence in the complexity, as in Table 1).
>
> That being said, we highlight that the comparison is not only "fair," but is exact; i.e., **the error budget for both methods are *exactly* prefixed**. Let us elaborate with Theorem 6 as an example.
>
> Theorem 6 decomposes the "improvement" and the "overhead" of our proposed meta algorithm, given --for example-- [1] as the input for QLSP_solver, which enjoys $\mathcal{O}(\kappa \cdot \log(\frac{1}{\varepsilon}))$ query complexity to acheieve $\varepsilon$-accurate solution for the QLSP problem in Definition 1. **The $\varepsilon$ appreaing on the RHS of the first inequality of Theorem 3 is *exactly* this $\varepsilon$.**
>
> Given the above, "wrapping" [1] with our proposed meta algorithm has two effects:
> - (improvement) the condition number changes to $\hat{\kappa}$ as in Eq (18), which is smaller than the original $\kappa$;
> - (overhead) to compensate, QLSP_solver should solve up to $\frac{\varepsilon}{c}$ accuracy for $c>1$.
>
> Figure 2 (left) shows how the "improvement" (blue line) is much more substantial for different values of $c$, compared to the the "overhead" (orangle line) which remains close-to-constant. This can be seen in Eq. (17) as $c$ enters a logarithmic term in the overhead.
>
> > Is there a practical way to choose the stepsize $\eta$ in the algorithm? What if the obtained solution is still far from the exact solution to the linear system?
>
> Thank you for the question. Indeed, $\eta$ is a hyperparameter, and according to our theory, it should be set to $\eta = \kappa \left( \frac{d}{\varepsilon_2} - 1 \right)$, which involves unknown quantities like $\kappa$ or $d = || x_0 - x^\star||$.
>
> However, we would like to remind about the "interpolation" detailed in Remark 3. The step size $\eta$ continuously interpolates the original problem (for $\eta = + \infty$) to a pre-conditioned problem (for small $\eta$), as the reviewer pointed out. That is, in practice, one can use "sufficiently large" $\eta$, and can only benefit from our meta algorithm.

---

> > ### Author Response · Authors · 2024-11-23
> >
> > > Is it possible to use the PPA solution as a "warm start" in a quantum linear system solver, which might be helpful to improve the overall performance?
> >
> > Thank you for the question. Indeed, as we mention above, using other method to "warm start" a quantum linear system solver is one of the main benefits our method provides, and is NOT possible with any other existing QLSP algorithm. In page 10 of the updated manuscript, we provide a simple illustration that, by using warm starting, one can roughly halve the total complexity, i.e., $\hat{\kappa} = \kappa/2$.
> >
> > > Can the authors further elaborate on how the comparison is made in Figure 1? Is it fair to include a $c$ parameter in Algorithm 1?
> >
> > Including $c$ makes it exact and fair. Notice that $c > 1$; our overall complexity in Figure 1, $\hat{\kappa} \log(c/\varepsilon)$, (i.e., LHS of Eq (17)) would even be smaller if we exclude $c$; this is exactly to prefix the $\varepsilon$ for both methods.
> >
> > That is, the output of the quantum algorithm from [1] (based on discrete adiabatic theorem) and the output of our meta algorithm with [1] as the QLSP_solver result in exact same final accuracy $\varepsilon$; but constant-level improvement in $\hat{\kappa} \leq \kappa$, which enters linearly in query complexity.
> >
> > We note that this is not a free lunch:
> > Figure 2 (left) shows how the "improvement" (blue line) is much more substantial for different values of $c$, compared to the the "overhead" (orangle line) which remains constant. This can be seen in Eq. (17) where $c$ enters a logarithmic term in the overhead.
> >
> >
> > We hope this clarifies your concerns. Please don't hesitate to follow up with unresolved concerns.
> >
> > [1] P. C.S. Costa, et al. (2022) "Optimal Scaling Quantum Linear-Systems Solver via Discrete Adiabatic Theorem"

---

> > > ### Comment · Reviewer_w21C · 2024-11-27
> > >
> > > We thank the authors for the detailed response. The explanations have effectively resolved my previous concerns, and I have updated my scores accordingly.

---

> > > > ### Author Response · Authors · 2024-11-27
> > > >
> > > > Dear Reviewer w21C,
> > > >
> > > > Thank you for the reply and updating the score accordingly.
> > > >
> > > > We want to point you to another update we made in the manuscript.
> > > > **In pg 10 (Figure 3)**, we included simulation results showcasing that **the optimal query complexity $\Omega(\kappa \log \frac{1}{\varepsilon})$ can effectively be halved via warm starting with gradient descent.**
> > > >
> > > > Specifically, we generate (normalized) $A$ and $b$ from $\mathcal{N}(0, 1)$, and vary the condition number of $A$ to be $\kappa \in \\{100, 200, 300, 400, 500\\}$. We plot the overall query complexity of our proposed method with warm start, where $x_0$ is initialized with the last iterate of $\\{200, 500, 1000\\}$ steps of gradient descent. One can see the overall query complexity gradually improves with better initialization, and can effectively halve the overall cost with 1000 steps of gradient descent. **We remind that "warm starting" or "better initialization" is NOT possible with any other existing QLSP solver.**
> > > >
> > > > We hope this addition clarifies your concern, and if so, we hope you could consider reflecting on the score. Thank you again for your time in reviewing our manuscript.

---

> > > > > ### Author Response · Authors · 2024-12-02
> > > > >
> > > > > Dear Reviewer w21C,
> > > > >
> > > > > Thank you again for your constructive feedback and thoughtful review.
> > > > > Given that the discussion period is ending soon, we would greatly appreciate if you could check out our **warm-starting result that we summarized in the above comment.**
> > > > >
> > > > > We hope this addition clarifies your concern, and if so, we hope you could consider reflecting on the score. Thank you again for your time in reviewing our manuscript.

---

> ### Comment · Area_Chair_sdc9 · 2024-11-27
>
> Dear Reviewer,
>
> The authors have provided their rebuttal to your comments/questions. Given that we are not far from the end of author-reviewer discussions, it will be very helpful if you can take a look at their rebuttal and provide any further comments. Even if you do not have further comments, please also confirm that you have read the rebuttal. Thanks!
>
> Best wishes,
> AC

---

### Official Review · Reviewer_RLwL · 2024-11-04

**Soundness:** 4
**Presentation:** 3
**Contribution:** 2
**Rating:** 5
**Confidence:** 4

**Summary:**

This paper investigates the quantum linear system problem (QLSP), which aims to produce a quantum state encoding the solution vector $x$ for a given matrix $A$ and vector $b$, such that $Ax = b$, assuming quantum query access to $A$ and $b$. The current best-known algorithm for QLSP achieves a query complexity of $O(\kappa \log \epsilon^{-1})$, where $\kappa$ is the condition number of $A$ and $\epsilon$ is the target error. While this result offers an exponential speedup in terms of $N$, the dimension of the system, the algorithm’s query complexity scales linearly with $\kappa$. Prior work has established that this linear dependence on $\kappa$ is unavoidable in the worst case, reducing the quantum advantage for poorly conditioned matrices.

This paper partially addresses this limitation by introducing a meta-algorithm for QLSP based on the proximal point algorithm (PPA)—a classical iterative optimization technique. This framework can be applied to any existing quantum linear system algorithm (QLSA) to modulate the tradeoff between solution precision and condition number. When applied to the current state-of-the-art QLSA, this meta-algorithm yields a constant-level improvement across a range of precision requirements, thereby extending the practical applicability of quantum linear system solvers.

**Strengths:**

This paper introduces a novel framework that enhances the dependence on the condition number $\kappa$ in quantum algorithms, while incorporating additional problem-dependent parameters. In the worst-case scenario, it achieves a significant constant-factor improvement over the existing state-of-the-art algorithm. The framework also includes a tunable parameter $\eta$, allowing users to adjust the balance between runtime and approximation error.

**Weaknesses:**

1. The meta algorithm itself is relatively simple, and the techniques used to analyze the problem is not extremely complicated.
2. The algorithm introduces additional problem-specific parameters, such as, $\Psi$ and $d\coloneqq ||x_0-x^*||$. Moreover, the paper did not provide a very thorough discussion on these parameters.

**Questions:**

1. Can the authors provide more discussions on these additionally introduced parameters?
2. Can the authors give a more detailed discussion on how the tradeoff between the runtime and the approximation error is achieved from the bound in Theorem 6?

---

> ### Author Response · Authors · 2024-11-23
>
> Thank you for your comments and feedback. We are pleased to hear that you found our approach novel, which indeed leads to a significant constant factor improvement.
>
> We address your comments and questions below in detail.
>
> > The meta algorithm itself is relatively simple, and the techniques used to analyze the problem is not extremely complicated.
>
> Thank you for the comment. We actually view this simplicity as one of the strengths of our work: by simply "wrapping" our meta-algorithm with a given QLSP_solver, one can achieve **a significant constant level improvement**, as you acknowledged in the strengths.
>
>
> > The algorithm introduces additional problem-specific parameters, such as, $\Psi$ and $d := || x_0 - x^\star ||$. Moreover, the paper did not provide a very thorough discussion on these parameters. Can the authors provide more discussions on these additionally introduced parameters?
>
> Thank you for the comment. We note that $\Psi := \sqrt{|| (I + \eta A)^{-1} (x_0 + \eta b)  || \cdot || A^{-1} b || }$ and $d := || x_0 - x^\star||$ indeed are additional parameters introduced.
>
> Intuitively, $\Psi \approx || x^\star || + \delta$. To see this, note that the first norm term within $\Psi$ can be analyzed as $|| (I + \eta A)^{-1} (x_0 + \eta b)  || \leq || (I + \eta A)^{-1} (x_0 + \eta b)  - A^{-1}b || + ||A^{-1} b\| \leq \varepsilon_2 + ||A^{-1} b||$ (c.f., Proposition 2). Thus, $\Psi = \sqrt{ (\varepsilon_2 + ||x^\star||) ||x^\star|| } \approx ||x^\star|| + \delta$. The other quantity $d := || x_0 - x^\star||$ is simply the distance to the solution from initialization $x_0$.
>
> These additional parameters precisely provide the significant constant-level speed-up of our meta algorithm. **Specifically, notice that both quantities depend on the step size $\eta$ and the initialization $x_0$, both of which are *tunable* by users**. In pg. 10 of our updated manuscript (teal color), we show how to halve the overall complexity of the current SOTA quantum algorithm [1] simply by using "warm up" (i.e., smarter initialization).
>
> This is impossible with other existing quantum algorithms; given $A$ and $b$, all existing quantum algorithms output quantum states proportional to $A^{-1}b$. Hence, if $A$ is ill-conditioned, existing quantum algorithms simply slow down due to (at best) linear dependence on $\kappa$ (c.f., Table 1), which can be improved (e.g., halved) by wrapping with our proposed meta-algorithm, agnostic of what QLSP_solver is used.
>
>
> > Can the authors give a more detailed discussion on how the tradeoff between the runtime and the approximation error is achieved from the bound in Theorem 6?
>
> Thank you for the question. Theorem 6 decomposes the "improvement" and the "overhead" of our proposed meta algorithm, given --for example-- [1] as the input for QLSP_solver, which enjoys $\mathcal{O}(\kappa \cdot \log(\frac{1}{\varepsilon}))$ query complexity to acheieve $\varepsilon$-accurate solution for the QLSP problem in Definition 1. We highlight that **the error budget $\varepsilon$ for both [1] and Algorithm 1 (with [1] as QLSP_solver) are *exactly* prefixed**.
>
> Now, "wrapping" [1] with our proposed meta algorithm has two effects:
> - (improvement) the condition number changes to $\hat{\kappa}$ as in Eq (18), which is smaller than the original $\kappa$;
> - (overhead) to compensate, QLSP_solver should solve up to $\frac{\varepsilon}{c}$ accuracy for $c>1$.
>
> Figure 2 (left) shows how the "improvement" (blue line) is much more substantial for different values of $c$, compared to the the "overhead" (orangle line) which remains close-to-constant. This can be seen in Eq. (17) as $c$ enters a logarithmic term in the overhead.
>
> We hope this clarifies your concerns. Please don't hesitate to follow up with unresolved concerns.
>
> [1] P. C.S. Costa, et al. (2022) "Optimal Scaling Quantum Linear-Systems Solver via Discrete Adiabatic Theorem"

---

> > ### Comment · Reviewer_RLwL · 2024-11-27
> >
> > Thank you for your detailed and thoughtful rebuttal. I truly appreciate the time and effort. Unfortunately, given the relatively large gap between a score of 5 and 6, I must respectfully maintain my original assessment.

---

> > > ### Author Response · Authors · 2024-11-27
> > >
> > > Dear Reviewer RLwL,
> > >
> > > Thank you for the reply. We respect your opinion.
> > >
> > > We still want to point you to another update we made in the manuscript.
> > > **In pg 10 (Figure 3)**, we included simulation results showcasing that **the optimal query complexity $\Omega(\kappa \log \frac{1}{\varepsilon})$ can effectively be halved via warm starting with gradient descent.**
> > >
> > > Specifically, we generate (normalized) $A$ and $b$ from $\mathcal{N}(0, 1)$, and vary the condition number of $A$ to be $\kappa \in \\{100, 200, 300, 400, 500\\}$. We plot the overall query complexity of our proposed method with warm start, where $x_0$ is initialized with the last iterate of $\\{200, 500, 1000\\}$ steps of gradient descent. One can see the overall query complexity gradually improves with better initialization, and can effectively halve the overall cost with 1000 steps of gradient descent. **We remind that "warm starting" or "better initialization" is NOT possible with any other existing QLSP solver.**
> > >
> > > We hope this addition clarifies your concern, and if so, we hope you could consider reflecting on the score. Thank you again for your time in reviewing our manuscript.

---

> > > > ### Author Response · Authors · 2024-12-02
> > > >
> > > > Dear Reviewer RLwL,
> > > >
> > > > Thank you again for your constructive feedback and thoughtful review.
> > > > Given that the discussion period is ending soon, we would greatly appreciate if you could check out our **warm-starting result that we summarized in the above comment.**
> > > >
> > > > We hope this addition clarifies your concern, and if so, we hope you could consider reflecting on the score. Thank you again for your time in reviewing our manuscript.

---

> ### Comment · Area_Chair_sdc9 · 2024-11-27
>
> Dear Reviewer,
>
> The authors have provided their rebuttal to your comments/questions. Given that we are not far from the end of author-reviewer discussions, it will be very helpful if you can take a look at their rebuttal and provide any further comments. Even if you do not have further comments, please also confirm that you have read the rebuttal. Thanks!
>
> Best wishes,
> AC

---

### Meta-Review · Area_Chair_sdc9 · 2024-12-07

**Metareview:**

This paper proposed a new framework for the quantum linear system problem via the proximal point problem. Specifically, this new framework can be seen as a meta-algorithm that adapts step size, and hence allows the user to control the trade-off between the runtime and the approximation error. Numerical experiments demonstrate the improvement of the proposed framework over previous algorithms.

During the review period, the reviewers acknowledged the contribution on having tunable parameters for the quantum linear system problem, and also benign numerical results. However, there are general concerns about the technical novelty over previous quantum computing results. Another main issue is that this paper focuses more on a technical problem in quantum computing - quantum linear system problem, but didn't introduce the impact to learning theory and potential applications. The scores are mostly on the reject side.

The final decision is hence rejection. For future versions, we encourage the authors to merge all points raised during the rebuttal to future versions of the paper. If the authors target for machine learning venues in the future, extension of the studied results to specific machine learning problems (that use solving linear systems) both in theory and experiments will be very appreciated.

**Additional Comments On Reviewer Discussion:**

During reviewer discussions, the authors adequately addressed issues raised by reviewers. Reviewer w21C updated score. However, the scores are mostly on the negative side, and reviewer eGnX who gave the score of 6 agreed with the concerns raised by other reviewers.

---

### Decision · Program_Chairs · 2025-01-22

Reject